# Prioritize Alignment in Dataset Distillation

.

## Abstract

Dataset Distillation aims to compress a large dataset into a significantly more compact, synthetic one without compromising the performance of the trained models. To achieve this, existing methods use the agent model to extract information from the target dataset and embed it into the distilled dataset. Consequently, the quality of extracted and embedded information determines the quality of the distilled dataset. In this work, we find that existing methods introduce misaligned information in both information extraction and embedding stages. To alleviate this, we propose Prioritize Alignment in Dataset Distillation (**PAD**), which aligns information from the following two perspectives. 1) We prune the target dataset according to the compressing ratio to filter the information that can be extracted by the agent model. 2) We use only deep layers of the agent model to perform the distillation to avoid excessively introducing low-level information. This simple strategy effectively filters out misaligned information and brings non-trivial improvement for mainstream matching-based distillation algorithms. Furthermore, built on trajectory matching, **PAD** achieves remarkable improvements on various benchmarks, achieving state-of-the-art performance. The code and distilled datasets will be made public.

## 1 Introduction

Dataset Distillation (DD) [43] aims to compress a large dataset into a small synthetic dataset that preserves important features for models to achieve comparable performances. Ever since being introduced, DD has gained a lot of attention because of its wide applications in practical fields such as privacy preservation [5, 44], continual learning [28, 35], and neural architecture search [12, 32].

Recently, matching-based methods [46, 42, 6] have achieved promising performance in distilling high-quality synthetic datasets. Generally, the process of these methods can be summarized into two steps: (1) *Information Extraction*: an agent model is used to extract important information from the target dataset by recording various metrics such as gradients [49], distributions [48], and training trajectories [1], (2) *Information Embedding*: the synthetic samples are optimized to incorporate the extracted information, which is achieved by minimizing the differences between the same metric calculated on the synthetic data and the one recorded in the previous step.

In this work, we first reveal both steps will introduce misaligned information, which is redundant and potentially detrimental to the quality of the synthetic data. Then, by analyzing the cause of this misalignment, we propose alleviating this problem through the following two perspectives.

Typically, in the *Information Extraction* step, most distillation methods allow the agent model to see all samples in the target dataset. This means information extracted by the agent model comes from samples with various difficulties (see Figure 1(a)). However, according to previous study

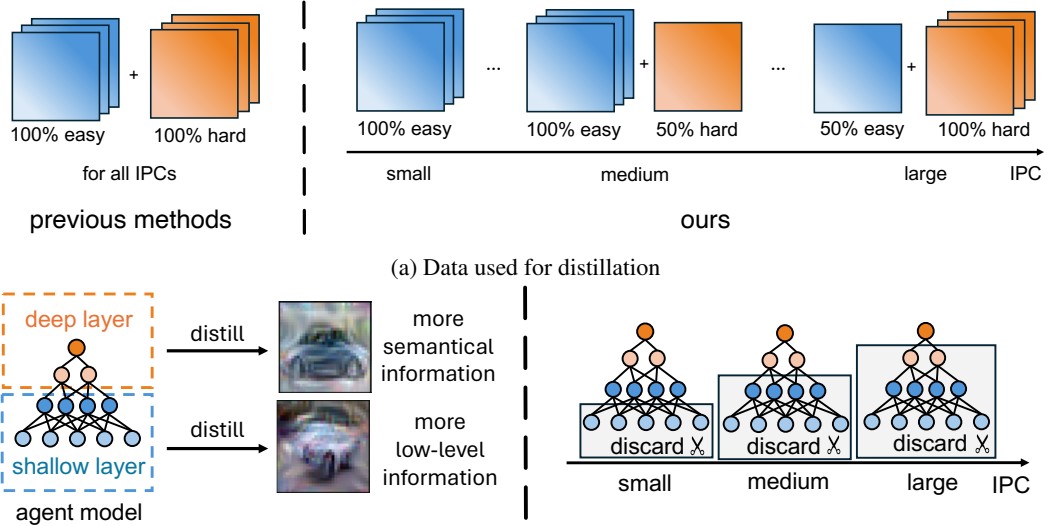

(a) Data used for distillation

(b) Parameters used for distillation

Figure 1: (a) Compared with using all samples without differentiation in IPCs (left), PAD meticulously selects a subset of samples for different IPCs to align the expected difficulty of information required (right). (b) Different layers distill different patterns (left). PAD masks out (grey box) shallow-layer parameters during metric matching in accordance with IPCs (right).

[10], information related to easy samples is only needed when the compression ratio is high. This misalignment leads to the sub-optimal of the distillation performance.

To alleviate the above issue, we first use data selection methods to measure the difficulty of each sample in the target dataset. Then, during the distillation, a data scheduler is employed to ensure only data whose difficulty is aligned with the compression ratio is available for the agent model.

In the *Information Embedding* step, most distillation methods except DM [48] choose to use all parameters of the agent model to perform the distillation. Intuitively, this will ensure the information extracted by the agent model is fully utilized. However, we find shallow layer parameters of the model can only provide low-quality, basic signals, which are redundant for dataset distillation in most cases. Conversely, performing the distillation with only parameters from deep layers will yield high-quality synthetic samples. We attribute this contradiction to the fact that deeper layers in DNNs tend to learn higher-level representations of input data [27, 37].

Based on our findings, to avoid embedding misaligned information in the *Information Embedding* step, we propose to use only parameters from deeper layers of the agent model to perform distillation, as illustrated in Figure 1(b). This simple change brings significant performance improvement, showing its effectiveness in aligning information.

Through experiments, we validate that our two-step alignment strategy is effective for distillation methods based on matching gradients [49], distributions [48], and trajectories [1]. Moreover, by applying our alignment strategy on trajectory matching [1, 10], we propose our novel method named Prioritize Alignment in Dataset Distillation (PAD). After conducting comprehensive evaluation experiments, we show PAD achieves state-of-the-art (SOTA) performance.

## 2 Misaligned Information in Dataset Distillation

Generally, we can summarize the distillation process of matching-based methods into the following two steps: (1) *Information Extraction*: use an agent model to extract essential information from the target dataset, realized by recording metrics such as gradients [49], distributions [48], and training trajectories [1], (2) *Information Embedding*: the synthetic samples are optimized to incorporate the extracted information, realized by minimizing the differences between the same metric calculated on the synthetic data and the one recorded in the first step.

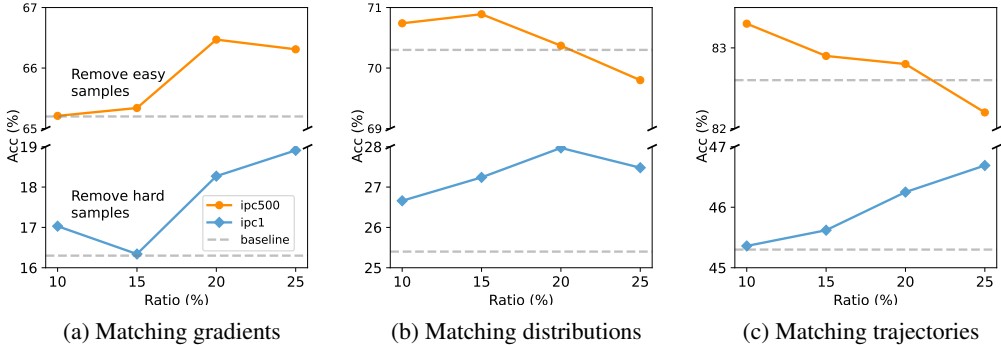

(a) Matching gradients      (b) Matching distributions      (c) Matching trajectories

Figure 2: Distillation performance on CIFAR-10 where data points are removed with different ratios. Removing unnecessary data points helps to improve the performance of methods based on matching gradients, distributions, and trajectories, both in low and high IPC cases.

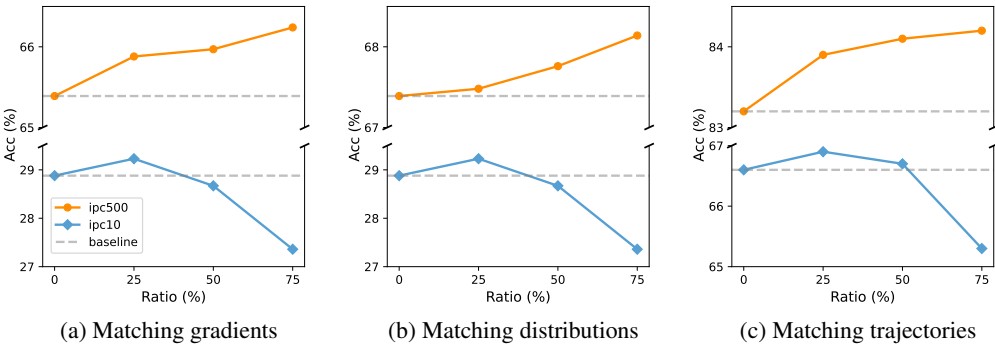

(a) Matching gradients      (b) Matching distributions      (c) Matching trajectories

Figure 3: Distillation performances on CIFAR-10 where n% (ratio) shallow layer parameters are not utilized during distillation. Discarding shallow-layer parameters is beneficial for methods based on matching gradients, distributions, and trajectories, both in low and high IPC cases.

In this section, through analyses and experimental verification, we show the above two steps both will introduce misaligned information to the synthetic data.

## 2.1 Misaligned Information Extracted by Agent Models

In the *information extraction* step, an agent model is employed to extract information from the target dataset. Generally, most existing methods [1, 6, 49, 46] allow the agent model to see the full dataset. This implies that the information extracted by the agent model originates from samples with diverse levels of difficulty. However, the expected difficulty of distilled information varies with changes in IPC: smaller IPCs prefer easier information while larger IPCs should distill harder one [10].

To verify if this misalignment will influence the quality of synthetic data, we perform the distillation where hard/easy samples of target dataset are removed with various ratios. As the results reported in Figure 2, pruning unaligned data points is beneficial for all matching-based methods. This proves the misalignment indeed will influence the distillation performance and can be alleviated by filtering out misaligned data from the target dataset.

## 2.2 Misaligned Information Embedded by Metric Matching

Most existing methods use all parameters of the agent model to compute the metric used for matching. Intuitively, this helps to improve the distillation performance, since in this way all information extracted by the agent model will be embedded into the synthetic dataset. However, since shallow layers in DNNs tend to learn basic distributions of data [27, 37], using parameters from these layers can only provide low-level signals that turned out to be redundant in most cases.

As can be observed in Figure 3, it is evident that across all matching-based methods, the removal of shallow layer parameters consistently enhances performance, regardless of the IPC setting. This proves employing over-shallow layer parameters to perform the distillation will introduce misaligned information to the synthetic data, compromising the quality of distilled data.

# 3 Method

To alleviate the information misalignment issue, based on trajectory matching (TM) [1, 10], we propose Prioritizing Alignment in Dataset Distillation (PAD). PAD can also be applied to methods based on matching gradients [49] and distributions [48], which are introduced in Appendix A.1.

## 3.1 Preliminary of Trajectory Matching

Following the two-step procedure, to extract information, TM-based methods [1, 10] first train agent models on the real dataset $\mathcal{D}_R$ and record the changes of the parameters. Specifically, let $\{\theta_t^*\}_0^N$ be an expert trajectory, which is a parameter sequence recorded during the training of agent model. At each iteration of trajectory matching, $\theta_t^*$ and $\theta_{t+M}^*$ are randomly selected from expert trajectories as the start and target parameters.

To embed the information into the synthetic data, TM methods minimize the distance between the expert trajectory and the student trajectory. Let $\hat{\theta}_t$ denote the parameters of the student agent model trained on synthetic dataset $\mathcal{D}_S$ at timestep $t$. The student trajectory progresses by doing gradient descent on the cross-entropy loss $l$ for $N$ steps:

$$\hat{\theta}_{t+i+1} = \hat{\theta}_{t+i} - \alpha \nabla l(\hat{\theta}_{t+i}, \mathcal{D}_S), \tag{1}$$

Finally, the synthetic data is optimized by minimizing the distance metric, which is formulated as:

$$\mathcal{L} = \frac{||\hat{\theta}_{t+N} - \theta_{t+M}^*||}{||\theta_{t+M}^* - \theta_t^*||}, \tag{2}$$

## 3.2 Filtering Information Extraction

In section 2.1, we show using data selection to filter out unmatched samples could alleviate the misalignment caused in *Information Extraction* step. According to previous work [10], TM-based methods prefer easy information and choose to match only early trajectories when IPC is small. Conversely, hard information is preferred by high IPCs and they match only late trajectories. Hence, we should use easy samples to train early trajectories, while late trajectories should be trained with hard samples. To realize this efficiently, we first use the data selection method to measure the difficulty of samples contained in the target dataset. Then, during training expert trajectories, a scheduler is implemented to gradually incorporate hard samples into the training set while excluding easier ones from it.

**Difficulty Scoring Function** Identifying the difficulty of data for DNNs to learn has been well studied in data selection area [29, 17, 16, 40]. For simplicity consideration, we use Error L2-Norm (EL2N) score [33] as the metric to evaluate the difficulty of training examples (other metrics can also be chosen, see Section 4.3.2). Specifically, let $x$ and $y$ denote a data point and its label, respectively. Then, the EL2N score can be calculated by:

$$\chi_t(x, y) = \mathbb{E}||p(w_t, x) - y||_2, \tag{3}$$

where $p(w_t, x) = \sigma(f(w_t, x))$ is the output of a model $f$ at training step $t$ transformed into a probability distribution. In consistent with [40], samples with higher EL2N scores are considered as harder samples in this paper.

**Scheduler** The scheduler can be divided into the following stages. Firstly, the hardest samples are removed from the training set, ensuring that it exclusively comprises data meeting a predetermined initial ratio (IR). Then, during training expert trajectories, samples are gradually added to the training set in order of increasing difficulty. After incorporating all the data into the training set, the scheduler will begin to remove easy samples from the target dataset. Unlike the gradual progression involved in adding data, the action of reducing data is completed in a single operation, since now the model has been trained on simple samples for a sufficient time.

| Dataset | CIFAR-10 | | | | | CIFAR-100 | | | | Tiny ImageNet | | |
|---|---|---|---|---|---|---|---|---|---|---|---|---|
| IPC | 1 | 10 | 50 | 500 | 1000 | 1 | 10 | 50 | 100 | 1 | 10 | 50 |
| Ratio | 0.02 | 0.2 | 1 | 10 | 20 | 0.2 | 2 | 10 | 20 | 0.2 | 2 | 10 |
| Random | 15.4±0.3 | 31.0±0.5 | 50.6±0.3 | 73.2±0.3 | 78.4±0.2 | 4.2±0.3 | 14.6±0.5 | 33.4±0.4 | 42.8±0.3 | 1.4±0.1 | 5.0±0.2 | 15.0±0.4 |
| KIP [31] | 49.9±0.2 | 62.7±0.3 | 68.6±0.2 | - | - | 15.7±0.2 | 28.3±0.1 | - | - | - | - | - |
| FRePo [50] | 46.8±0.7 | 65.5±0.4 | 71.7±0.2 | - | - | 28.7±0.1 | 42.5±0.2 | 44.3±0.2 | - | 15.4±0.3 | 25.4±0.2 | - |
| RCIG [26] | 53.9±1.0 | 69.1±0.4 | 73.5±0.3 | - | - | 39.3±0.4 | 44.1±0.4 | 46.7±0.3 | - | 25.6±0.3 | 29.4±0.2 | - |
| DC [49] | 28.3±0.5 | 44.9±0.5 | 53.9±0.5 | 72.1±0.4 | 76.6±0.3 | 12.8±0.3 | 25.2±0.3 | - | - | - | - | - |
| DM [48] | 26.0±0.8 | 48.9±0.6 | 63.0±0.4 | 75.1±0.3 | 78.8±0.1 | 11.4±0.3 | 29.7±0.3 | 43.6±0.4 | - | 3.9±0.2 | 12.9±0.4 | 24.1±0.3 |
| DSA [47] | 28.8±0.7 | 52.1±0.5 | 60.6±0.5 | 73.6±0.3 | 78.7±0.3 | 13.9±0.3 | 32.3±0.3 | 42.8±0.4 | - | - | - | - |
| TESLA [4] | **48.5±0.8** | 66.4±0.8 | 72.6±0.7 | - | - | 24.8±0.4 | 41.7±0.3 | 47.9±0.3 | 49.2±0.4 | - | - | - |
| CAFE [42] | 30.3±1.1 | 46.3±0.6 | 55.5±0.6 | - | - | 12.9±0.3 | 27.8±0.3 | 37.9±0.3 | - | - | - | - |
| MTT [1] | 46.2±0.8 | 65.4±0.7 | 71.6±0.2 | - | - | 24.3±0.3 | 39.7±0.4 | 47.7±0.2 | 49.2±0.4 | 8.8±0.3 | 23.2±0.2 | 28.0±0.3 |
| FTD [6] | 46.0±0.4 | 65.3±0.4 | 73.2±0.2 | - | - | 24.4±0.4 | 42.5±0.2 | 48.5±0.3 | 49.7±0.4 | 10.5±0.2 | 23.4±0.3 | 28.2±0.4 |
| DATM [10] | 46.9±0.5 | 66.8±0.2 | 76.1±0.3 | 83.5±0.2 | 85.5±0.4 | 27.9±0.2 | 47.2±0.4 | 55.0±0.2 | 57.5±0.2 | 17.1±0.3 | 31.1±0.3 | 39.7±0.3 |
| **PAD** | 47.2±0.6 | **67.4±0.3** | **77.0±0.5** | **84.6±0.3** | **86.7±0.2** | **28.4±0.5** | **47.8±0.2** | **55.9±0.3** | **58.5±0.3** | **17.7±0.2** | **32.3±0.4** | **41.6±0.4** |
| Full Dataset | | 84.8±0.1 | | | | | 56.2±0.3 | | | | 37.6±0.4 | |

Table 1: Comparison with previous dataset distillation methods (bottom: matching-based, top: others) on CIFAR-10, CIFAR-100 and Tiny ImageNet. ConvNet is used for the distillation and evaluation. Our method consistently outperforms prior matching-based methods.

### 3.3 Filtering Information Embedding

To filter out misaligned information introduced by matching shallow-layer parameters, we propose to add a parameter selection module that masks out part of shallow layers for metric computation. Specifically, parameters of an agent network can be represented as a flattened array of length $L$ that stores weights of agent models ordered from shallow to deep layers (parameters within the same layer are sorted in default order). The parameter selection sets a threshold ratio $\alpha$ such that the first $k = L \cdot \alpha$ parameters are not used for distillation. Then the parameters used for matching can now be formulated as:

$$\hat{\theta}_{t+N} = \{\underbrace{\hat{\theta}_0, \hat{\theta}_1, \cdots, \hat{\theta}_{k-1}}_{\text{discard}}, \underbrace{\hat{\theta}_k, \hat{\theta}_{k+1}, \cdots, \hat{\theta}_L}_{\text{used for matching}}\}. \tag{4}$$

In practice, the ratio $\alpha$ should vary with the change of IPC. For smaller IPCs, it is necessary to incorporate basic information thus $\alpha$ should be lower. Conversely, basic information is redundant in larger IPC cases, so $\alpha$ should be higher accordingly.

## 4 Experiments

### 4.1 Settings

We compare PAD with several prominent dataset distillation methods, which can be divided into two categories: matching-based approaches including DC [49], DM [48], DSA [47], CAFE [42], MTT [1], FTD [6], DATM [10], TESLA [4], and kernel-based approaches including KIP [31], FRePo [50], RCIG [26]. The assessment is conducted on widely recognized datasets: CIFAR-10, CIFAR-100[18], and Tiny ImageNet [20]. We implemented our method based on DATM [10]. In both the distillation and evaluation phases, we apply the standard set of differentiable augmentations commonly used in previous studies [1, 6, 10]. By default, networks are constructed with instance normalization unless explicitly labeled with "-BN," indicating batch normalization (e.g., ConvNet-BN). For CIFAR-10 and CIFAR-100, distillation is typically performed using a 3-layer ConvNet, while Tiny ImageNet requires a 4-layer ConvNet. Cross-architecture experiments also utilize LeNet [21], AlexNet [19], VGG11 [39], and ResNet18 [11]. More details can be found in the appendix.

### 4.2 Main Results

**CIFAR and Tiny ImageNet** We conduct comprehensive experiments to compare the performance of our method with previous works. As the results presented in Table 1, PAD outperforms previous matching-based methods on three datasets except for the case when IPC=1. When compared with kernel-based methods which use a larger network to perform the distillation, our technique exhibits superior performance in most cases, particularly when the compression ratio exceeds 1%. As can be observed, PAD performs relatively better when IPC is high, suggesting our filtering out misaligned information strategy becomes increasingly effective as IPC increases.

| Dataset | Ratio | Method | ConvNet | ConvNet-BN | ResNet18 | ResNet18-BN | VGG11 | AlexNet | LeNet | MLP | Avg. |
|---|---|---|---|---|---|---|---|---|---|---|---|
| CIFAR-10 | 20% | Random | 78.38 | 80.25 | 84.58 | 87.21 | 80.81 | 80.75 | 61.85 | 50.98 | 75.60 |
| | | Glister | 62.46 | 70.52 | 81.10 | 74.59 | 78.07 | 70.55 | 56.56 | 40.59 | 66.81 |
| | | Forgetting | 76.27 | 80.06 | 85.67 | 87.18 | 82.04 | 81.35 | 64.59 | 52.21 | 76.17 |
| | | DATM | 85.50 | 85.23 | **87.22** | **88.13** | **84.65** | 85.14 | 66.70 | 52.40 | 79.37 |
| | | **PAD** | **86.90** | **85.67** | 86.95 | 88.09 | 84.34 | **85.83** | **67.28** | **53.62** | **79.84** |
| | | ↑ | +8.52 | +5.42 | +2.37 | +0.88 | +3.53 | +5.08 | +5.43 | +2.64 | +4.24 |
| CIFAR-100 | 20% | Random | 42.80 | 46.38 | 47.48 | 55.62 | 42.69 | 38.05 | 25.91 | 20.66 | 39.95 |
| | | Glister | 35.45 | 37.13 | 42.49 | 46.14 | 43.06 | 28.58 | 23.33 | 17.08 | 34.16 |
| | | Forgetting | 45.52 | 49.99 | 51.44 | 54.65 | 43.28 | 43.47 | 27.22 | 22.90 | 42.30 |
| | | DATM | 57.50 | 57.75 | 57.98 | **63.34** | **55.10** | 55.69 | 33.57 | 26.39 | 50.92 |
| | | **PAD** | **58.50** | **58.66** | **58.15** | 63.17 | 55.02 | **55.93** | **33.87** | **27.12** | **51.30** |
| | | ↑ | +15.70 | +12.28 | +10.67 | +7.55 | +12.33 | +17.88 | +7.96 | +6.46 | +11.35 |
| Tiny | 10% | Random | 15.00 | 24.21 | 17.73 | 28.07 | 22.51 | 14.03 | 9.25 | 5.85 | 17.08 |
| | | Glister | 17.32 | 19.77 | 18.84 | 23.12 | 19.10 | 11.68 | 8.84 | 3.86 | 15.32 |
| | | Forgetting | 20.04 | 23.83 | 19.38 | 28.88 | 23.77 | 12.13 | 12.06 | 5.54 | 18.20 |
| | | DATM | 39.68 | 40.32 | **36.12** | **43.14** | 38.35 | **35.10** | 12.41 | 9.02 | 31.76 |
| | | **PAD** | **41.02** | **40.88** | 36.08 | 42.96 | **38.64** | 35.02 | **13.17** | **9.68** | **32.18** |
| | | ↑ | +26.02 | +16.67 | +18.35 | +14.89 | +16.13 | +20.99 | +3.92 | +3.83 | +15.10 |

Table 2: Cross-architecture evaluation of distilled data on unseen networks. Results worse than random selection are indicated with red color. ↑ denotes the performance improvement brought by our method compared with random selection. Tiny denotes Tiny ImageNet.

| Method | ConvNet | ResNet18 | VGG | AlexNet |
|---|---|---|---|---|
| Random | 33.46 | 31.95 | 32.18 | 26.65 |
| FTD | 48.90 | 46.65 | 43.24 | 42.20 |
| DATM | 55.03 | 51.71 | **45.38** | 45.74 |
| **PAD** | **55.91** | **52.35** | 44.97 | **45.92** |

| FIEX | FIEM | Accuracy(%) |
|---|---|---|
| | | 66.7 |
| | ✓ | 66.9 |
| ✓ | | 67.2 |
| ✓ | ✓ | 67.4 |

| IR | AEE | | |
|---|---|---|---|
| | 20 | 40 | 60 |
| 50% | 66.23 | 66.07 | 65.92 |
| 75% | **67.36** | **67.34** | **66.58** |
| 80% | 67.26 | 67.08 | 66.47 |

(a) Datasets distilled by PAD generalize well across various architectures.

(b) Each module brings non-trivial improvements.

(c) Set IR as 75% always perform best.

Table 3: **(a)** Cross-Architecture evaluation on CIFAR-100 IPC50. **(b)** Ablation studies on the modules of our method on CIFAR-10 IPC10. **(c)** Results of different sets of data selection hyper-parameters on CIFAR-10 IPC10.

**Cross Architecture Generalization** We evaluate the generalizability of our distilled data in both low and high IPC cases. As results reported in Table 3(a), when IPC is small, our distilled data outperforms the previous SOTA method DATM on ResNet and AlexNet while maintaining comparable accuracy on VGG. This suggests that our distilled data on high compressing ratios generalizes well across various unseen networks. Moreover, as reflected in Table 2, our distilled datasets on large IPCs also have the best performance on most evaluated architectures, showing good generalizability in the low compressing ratio case.

## 4.3 Ablation Study

To validate the effectiveness of each component of our method, we conducted ablation experiments on modules (section 4.3.1) and their hyper-parameter settings (section 4.3.2 and section 4.3.2).

### 4.3.1 Modules

Our method incorporates two separate modules to filter information extraction (FIEX) and information embedding (FIEM), respectively. To verify their isolated effectiveness, we conduct an ablation study by applying two modules individually. As depicted in Table 3(b), both FIEX and FIEM bring improvements, implying their efficacy. By applying these two modules, we are able to effectively remove unaligned information, improving the distillation performance.

### 4.3.2 Hyper-parameters of Filtering Information Extraction

**Initial Ratio and Data Addition Epoch** To filter the information learned by agent models, we initialize the training set with only easy samples, and the size is determined by a certain ratio of the total size. Then, we gradually add hard samples into the training set. In practice, we use two hyper-parameters to control the addition process: the initial ratio (IR) of training data for training set initialization and the end epoch of hard sample addition (AEE). These two parameters together control the amount of data agent models can see at each epoch and the speed of adding hard samples.

| Method | IPC | | |
|---|---|---|---|
| | 1 | 10 | 500 |
| Loss | 45.74 | 66.45 | 83.47 |
| Uncertainty [3] | 46.22 | 66.99 | 84.22 |
| EL2N [33] | **47.23** | **67.38** | **84.63** |

| IPC | Ratio | | | |
|---|---|---|---|---|
| | 100% | 75% | 50% | 25% |
| 1 | **47.2** | 46.56 | 45.98 | 41.32 |
| 10 | 67.2 | **67.34** | 66.86 | 65.15 |
| 500 | 83.71 | 83.82 | **84.23** | 84.64 |

| Strategy | IPC | |
|---|---|---|
| | 10 | 50 |
| Baseline | 67.2 | 76.5 |
| Loss | 67.3 | 77.0 |
| Depth | **67.7** | **77.3** |

(a) Using EL2N to measure the difficulty of samples has the best performance.

(b) As IPC increases, removing more shallow-layer parameters becomes more effective.

(c) Using layer depth to select parameters outperforms using matching loss.

Table 4: **(a)** Ablation of different difficulty scoring functions on CIFAR-10. **(b)** Results of masking out different ratios of shallow-layer parameters across various IPCs on CIFAR-10. **(c)** Ablation on the strategy used for parameter selection on CIFAR-10

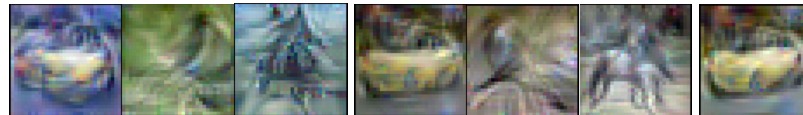

(a) with 100% parameters      (b) with 75% parameters      (c) with 50% parameters

Figure 4: Synthetic images of CIFAR-10 IPC50 obtained by PAD with different ratios of parameter selection. Smoother image features indicate that by removing some shallow-layer parameters during matching, PAD successfully filters out coarse-grained low-level information.

In Table 3(c), we show the distillation results where different hyper-parameters are utilized. In general, a larger initial ratio and faster speed of addition bring better performances. Although the distillation benefited more from learning simpler information when IPC is small [10], our findings indicate that excessively removing difficult samples (e.g., more than a quarter) early in the training phase can adversely affect the distilled data. This negative impact is likely due to the excessive removal leading to distorted feature distributions within each category. On the other hand, reasonably improving the speed of adding hard samples allows the agent model to achieve a more balanced learning of information of varying difficulty across different stages.

**Other Difficulty Scoring Functions**    Identifying the difficulty of data points is the key to filtering out misaligned information in the extraction step. Here, we compare the effect of using other difficulty-scoring functions to evaluate the difficulty of data. (1) prediction loss of a pre-trained ResNet. (2) uncertainty score [3]. (3) EL2N [33]. As can be observed in Table 4(a), EL2N performs the best across various IPCs; thus, we use it to measure how hard each data point is as default in our method. Note that this can also be replaced with a more advanced data selection algorithm.

### 4.3.3   Ratios of Parameter Selection

It is important to find a good balance between the percentage of shallow-layer parameters removed from matching and the loss of information. In Table 4(b), we show results obtained on different IPCs by discarding various ratios of shallow-layer parameters. The impact of removing varying proportions of shallow parameters on the distilled data and its relationship with changes in IPC is consistent with prior conclusions. For small IPCs, distilled data requires more low-level basic information. Thus, removing too many shallow-layer parameters causes a negative effect on the classification performance. By contrast, high-level semantic information is more important when it comes to large IPCs. With increasing ratios of shallow-layer parameters being discarded, we can ensure that low-level information is effectively filtered out from the distilled data.

## 5   Discussion

### 5.1   Distilled Images with Filtering Information Embedding

To see the concrete patterns brought by removing shallow-layer parameters to perform the trajectory matching, we present distilled images obtained by discarding various ratios of shallow-layer parameters in Figure 4. As can be observed in Figure 4(a), without removing any shallow-layer parameters

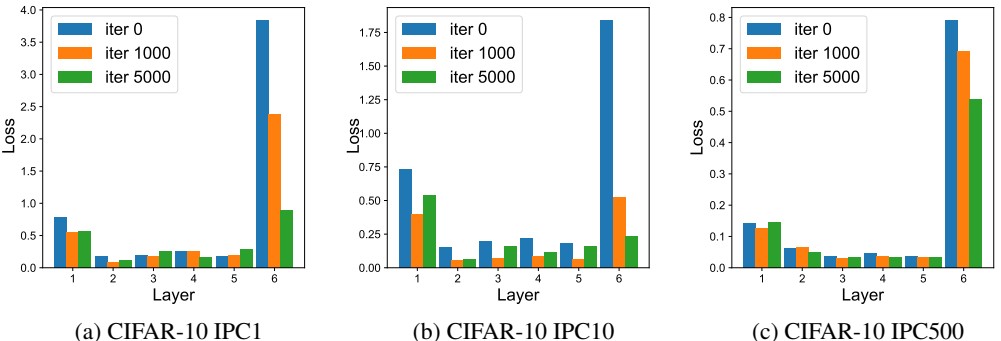

(a) CIFAR-10 IPC1 (b) CIFAR-10 IPC10 (c) CIFAR-10 IPC500

Figure 5: Losses of different layers of ConvNet after matching trajectories for 0, 1000, and 5000 iterations. We notice a similar phenomenon on both small (IPC1 and IPC10) and large IPCs (IPC500): losses of shallow-layer parameters fluctuate along the matching process, while losses of deep-layer parameters show a clear trend of decreasing.

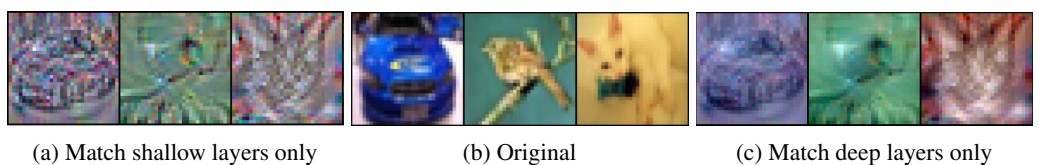

(a) Match shallow layers only (b) Original (c) Match deep layers only

Figure 6: Synthetic images visualization with parameter selection. Matching parameters in shallow layers produces an abundance of low-level texture features, whereas patterns generated by matching deep-layer parameters embody richer high-level semantic information.

to filter misaligned information, synthetic images are interspersed with substantial noises. These noises often take the form of coarse and generic information, such as the overall color distribution and edges in the image, which provides minimal utility for precise classification.

By contrast, images distilled by our enhanced methodology (see Figure 4(b) and Figure 4(c)), which includes meticulous masking out shallow-layer parameters during trajectory matching according to the compressing ratio, contain more fine-grained and smoother features. These images also encapsulate a broader range of semantic information, which is crucial for helping the model make accurate classifications. Moreover, we observe a clear trend: as the amount of the removed shallow-layer parameters increases, the distilled images exhibit clearer and smoother features.

## 5.2 Rationale for Parameter Selection

In this section, we analyze from the perspective of trajectory matching why shallow-layer parameters should be masked out. In Figure 5, we present the changes in trajectory matching loss across different layers as the distillation progresses. Compared to the deep-layer parameters of the agent model, a substantial number of shallow-layer parameters exhibit low loss values that fluctuate during the matching process (see Figure 5). By contrast, the loss values of the deep layers are much higher but consistently decrease as distillation continues. This suggests that matching shallow layers primarily conveys low-level information that is readily captured by the synthetic data and quickly saturated. Consequently, the excessive addition of such low-level information produces noise, reducing the quality of distilled datasets.

For a concrete visualization, we provide distilled images resulting from using only shallow-layer parameters or only deep-layer parameters to match trajectories in Figure 6. The coarse image features depicted in Figure 6(a) further substantiate our analysis.

## 5.3 Parameter Selection Strategy

In the previous section, we observed a positive correlation between the depth of the model layers and the magnitude of their trajectory-matching losses. Notably, the loss in the first layer of the ConvNet was higher compared to other shallow layers. Consequently, we further compared different parameter alignment strategies, specifically by sorting the parameters based on their matching losses and excluding a certain proportion of parameters with lower losses. Higher loss values indicate greater discrepancies in parameter weights; thus, continuing to match these parameters can inject more information into the synthetic data. As shown in Table 4(c), sorting by loss results in an improvement compared with no parameter alignment, but filtering based on parameter depth proves to be more effective.

## 6  Related Work

Introduced by [43], dataset distillation aims to synthesize a compact set of data that allows models to achieve similar test performances compared with the original dataset. Since then, a number of studies have explored various approaches. These methods can be divided into three types: kernel-based, matching-based, and using generative models [45].

**Kernel-based methods** are able to achieve closed-form solutions for the inner optimization [31] via kernel ridge regression with NTK [22]. FRePo [50] distills a compact dataset through neural feature regression and reduces the training cost.

**Matching-based methods** first use agent models to extract information from the target dataset by recording a specific metric [7, 23, 38, 24]. Representative works that design different metrics include DC [49] that matches gradients, DM [48] that matches distributions, and MTT [1] that matches training trajectories. Then, the distilled dataset is optimized by minimizing the matched distance between the metric computed on synthetic data and the record one from the previous step. Following this workflow, many works have been proposed to improve the efficacy of the distilled dataset. For example, CAFE [42] preserves the real feature distribution and the discriminative power of the synthetic data and achieves prominent generalization ability across various architectures. DREAM [25] employs K-Means to select representative samples for distillation and improves the distillation efficiency. DATM [10] proposes to match early trajectories for small IPCs and late trajectories for large IPCs, achieving SOTA performances on several benchmarks. Moreover, new metrics such as spatial attention maps [36, 15] have also been introduced and achieved promising performance in distilling large-scale datasets.

**Generative models** such as GANs [8, 13, 14, 41] and diffusion models [34, 30, 9] can also be used to distill high quality datasets. DiM [41] uses deep generative models to store information of the target dataset. GLaD [2] transfers synthetic data optimization from the pixel space to the latent space by employing deep generative priors. It enhances the generalizability of previous distillation methods.

## 7  Conclusion

In this work, we find a limitation of existing Dataset Distillation methods in that they will introduce misaligned information to the distilled datasets. To alleviate this, we propose PAD, which incorporates two modules to filter out misaligned information. For information extraction, PAD prunes the target dataset based on sample difficulty for different IPCs so that only information with aligned difficulty is extracted by the agent model. For information embedding, PAD discards part of shallow-layer parameters to avoid injecting low-level basic information into the synthetic data. PAD achieves SOTA performance on various benchmarks. Moreover, we show PAD can also be applied to methods based on matching gradients and distribution, bringing remarkable improvements across various IPC settings.

**Limitations**   Our alignment strategy could also be applied to methods based on matching gradients and distributions (see Appendix A.1). However, due to the limitation of computing resources, for methods based on matching distributions and gradients, we have only validated our method's effectiveness on DM [48] and DC [49] (see Table 5 and Table 6).

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

| IPC | Ratio | | | | Baseline |
|-----|-------|-------|-------|-------|----------|
|     | 10%   | 15%   | 20%   | 25%   |          |
| 1   | 17.03 | 16.34 | 18.27 | **18.91** | 16.32 |
| 500 | 65.21 | 65.34 | **66.47** | 66.31 | 65.27 |

(a) Removing various ratios of hard/easy samples improves DC on small/large IPCs.

| IPC | Ratio | | | | Baseline |
|-----|-------|-------|-------|-------|----------|
|     | 10%   | 15%   | 20%   | 25%   |          |
| 1   | 26.66 | 27.24 | **27.97** | 27.48 | 25.41 |
| 500 | 70.74 | **70.89** | 70.37 | 69.80 | 70.32 |

(b) Removing various ratios of hard/easy samples improves DM on small/large IPCs.

Table 5: Results of filtering information extraction by removing hard/easy samples in DC(a) and DM(b) on CIFAR-10.

| IPC | Ratio | | | Baseline |
|-----|-------|-------|-------|----------|
|     | 25%   | 50%   | 75%   |          |
| 10  | **29.23** | 28.67 | 27.36 | 28.88 |
| 500 | 65.88 | 65.97 | **66.24** | 65.39 |

(a) Matching gradients from deep-layer parameters leads to improvements.

| IPC | Ratio | | | Baseline |
|-----|-------|-------|-------|----------|
|     | 25%   | 50%   | 75%   |          |
| 10  | **29.23** | 28.67 | 27.36 | 28.88 |
| 500 | 67.48 | 67.76 | **68.14** | 67.39 |

(b) Matching distributions from deep-layer parameters leads to improvements.

Table 6: Results of filtering information embedding by masking out shallow-layer parameters for metric computation in DC(a) and DM(b) on CIFAR-10.

# A  Appendix

## A.1  Filtering Misaligned Information in DC and DM

Although PAD is implemented based on trajectory matching methods, we also test our proposed data alignment and parameter alignment on gradient matching and distribution matching. The performances of enhanced DC and DM with each of the two modules are reported in Table 5 and Tabl 6, respectively. We provide details of how we integrate these two modules into gradient matching and distribution matching in the following sections.

**Gradient Matching** We use the official implementation[1] of DC [49]. In the Information Extraction step, DC uses an agent model to calculate the gradients after being trained on the target dataset. We employ filter misaligned information in this step as follows: When IPC is small, a certain ratio of hard samples is removed from the target dataset so that the recorded gradients only contain simple information. Conversely, when IPC becomes large, we remove easy samples instead.

In the Information Embedding step, DC optimizes the synthetic data by back-propagating on the gradient matching loss. The loss is computed by summing the differences in gradients between each pair of model parameters. Thus, we apply parameter selection by discarding a certain ratio of parameters in the shallow layers.

**Distribution Matching** We use the official implementation of DM [48], which can be accessed via the same link as DC. In the Information Extraction step, DM uses an agent model to generate embeddings of input images from the target dataset. Similarly, filtering information extraction is applied by removing hard samples for small IPCs and easy samples for large IPCs.

In the Information Embedding step, since DM only uses the output of the last layer to match distributions, we modify the implementation of the network such that outputs of each layer in the model are returned by the forward function. Then, we perform parameter selection following the same practice as before.

## A.2  Experiment Settings

We use DATM [10] as the backbone TM algorithm and our proposed PAD is built upon. Thus, our configurations for distillation, evaluation, and network are consistent with DATM.

---

[1]https://github.com/VICO-UoE/DatasetCondensation.git

434 **Distillation.** We conduct the distillation process for 10,000 iterations to ensure full convergence of
435 the optimization. By default, ZCA whitening is applied in all the experiments.

436 **Evaluation.** We train a randomly initialized network on the distilled dataset and evaluate its per-
437 formance on the entire validation set of the original dataset. Following DATM [10], the evaluation
438 networks are trained for 1000 epochs to ensure full optimization convergence. For fairness, the
439 experimental results of previous distillation methods in both low and high IPC settings are sourced
440 from [10].

441 **Network.** We employ a range of networks to assess the generalizability of our distilled datasets.
442 For scaling ResNet, LeNet, and AlexNet to Tiny-ImageNet, we modify the stride of their initial
443 convolutional layer from 1 to 2. In the case of VGG, we adjust the stride of its final max pooling
444 layer from 1 to 2. The MLP used in our evaluations features a single hidden layer with 128 units.

445 **Hyper-parameters.** Hyper-parameters of our experiments on CIFAR-10, CIFAR-100, and Tiny-
446 ImageNet are reported in Table 7. Hyper-parameters can be divided into three parts including data
447 alignment (DA), parameter alignment (PA) and trajectory matching (TM). Soft labels are applied in
448 all experiments , we set its momentum to 0.9.

449 **Compute resources.** Our experiments are run on 4 NVIDIA A100 GPUs, each with 80 GB of
450 memory. The amount of GPU memory needed is mainly determined by the batch size of synthetic
451 data and the number of steps that the agment model is trained on synthetic data. To reduce the GPU
452 usage when IPC is large, one can apply TESLA [4] or simply reducing the synthetic steps $N$ or the
453 synthetic batch size. However, the decrement of hyper-parameters shown in Table 7 could result in
454 performance degradation.

| Dataset | IPC | DA | | PA | | | | | | | TM | | | |
| | | IR | AEE | $\alpha$ | N | M | $T^-$ | $T$ | $T^+$ | Interval | Synthetic Batch Size | Learning Rate (Label) | Learning Rate (Pixels) |
|---|---|---|---|---|---|---|---|---|---|---|---|---|---|
| CIFAR-10 | 1 | | | 0% | 80 | 2 | 0 | 4 | 4 | - | 10 | 5 | 100 |
| | 10 | | | 25% | 80 | 2 | 0 | 10 | 20 | 100 | 100 | 2 | 100 |
| | 50 | 0.75 | 20 | 25% | 80 | 2 | 0 | 20 | 40 | 100 | 500 | 2 | 1000 |
| | 500 | | | 50% | 80 | 2 | 40 | 60 | 60 | - | 1000 | 10 | 50 |
| | 1000 | | | 75% | 80 | 2 | 40 | 60 | 60 | - | 1000 | 10 | 50 |
| CIFAR-100 | 1 | | | 0% | 40 | 3 | 0 | 10 | 20 | 100 | 100 | 10 | 1000 |
| | 10 | 0.75 | 40 | 25% | 80 | 2 | 0 | 20 | 40 | 100 | 1000 | 10 | 1000 |
| | 50 | | | 50% | 80 | 2 | 40 | 60 | 80 | 100 | 1000 | 10 | 1000 |
| | 100 | | | 50% | 80 | 2 | 40 | 80 | 80 | - | 1000 | 10 | 50 |
| TI | 1 | | | 0% | 60 | 2 | 0 | 15 | 30 | 400 | 200 | 10 | 10000 |
| | 10 | 0.75 | 40 | 25% | 60 | 2 | 0 | 20 | 40 | 100 | 250 | 10 | 100 |
| | 50 | | | 50% | 80 | 2 | 20 | 40 | 60 | 100 | 250 | 10 | 100 |

Table 7: Hyper-parameters for different benchmarks.

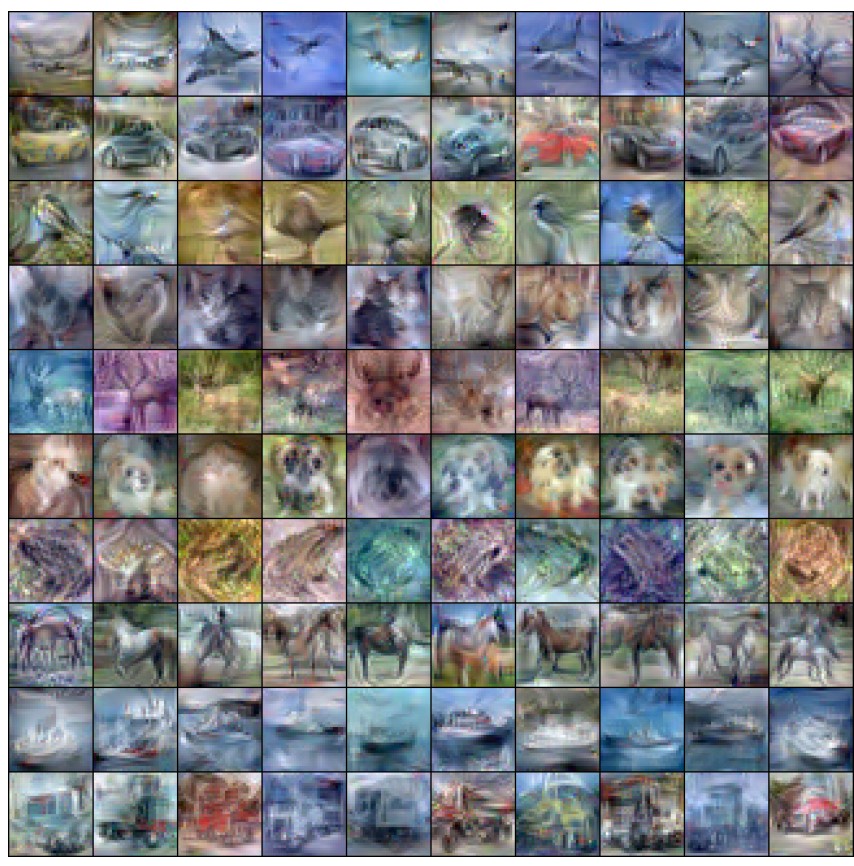

Figure 7: Distilled images of CIFAR-10 IPC10

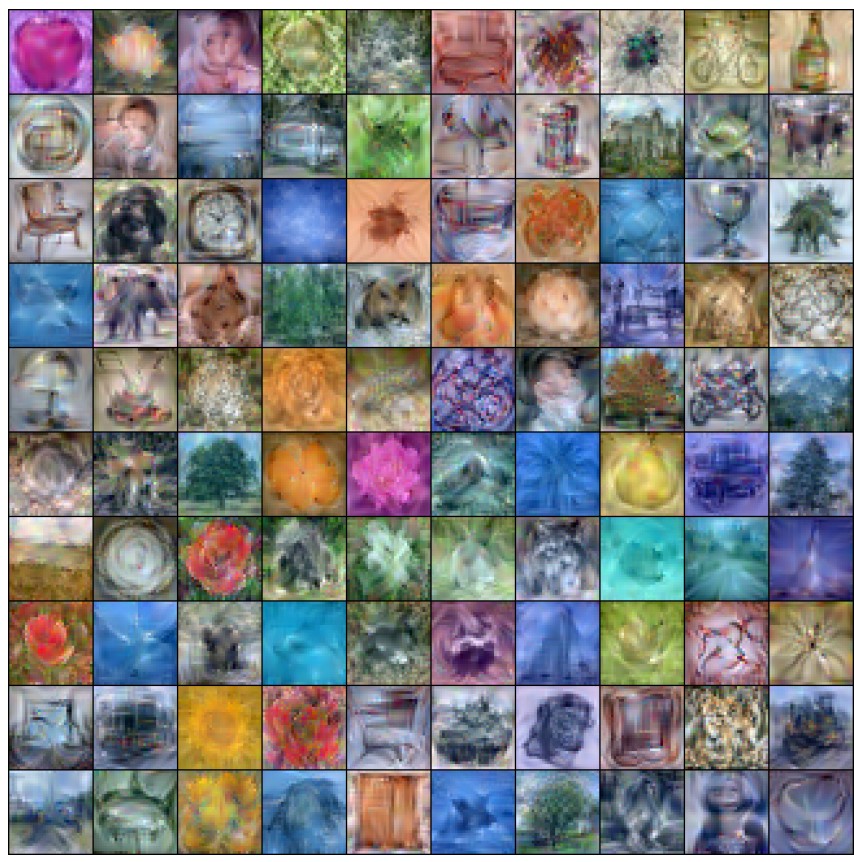

Figure 8: Distilled images of CIFAR-10 IPC10

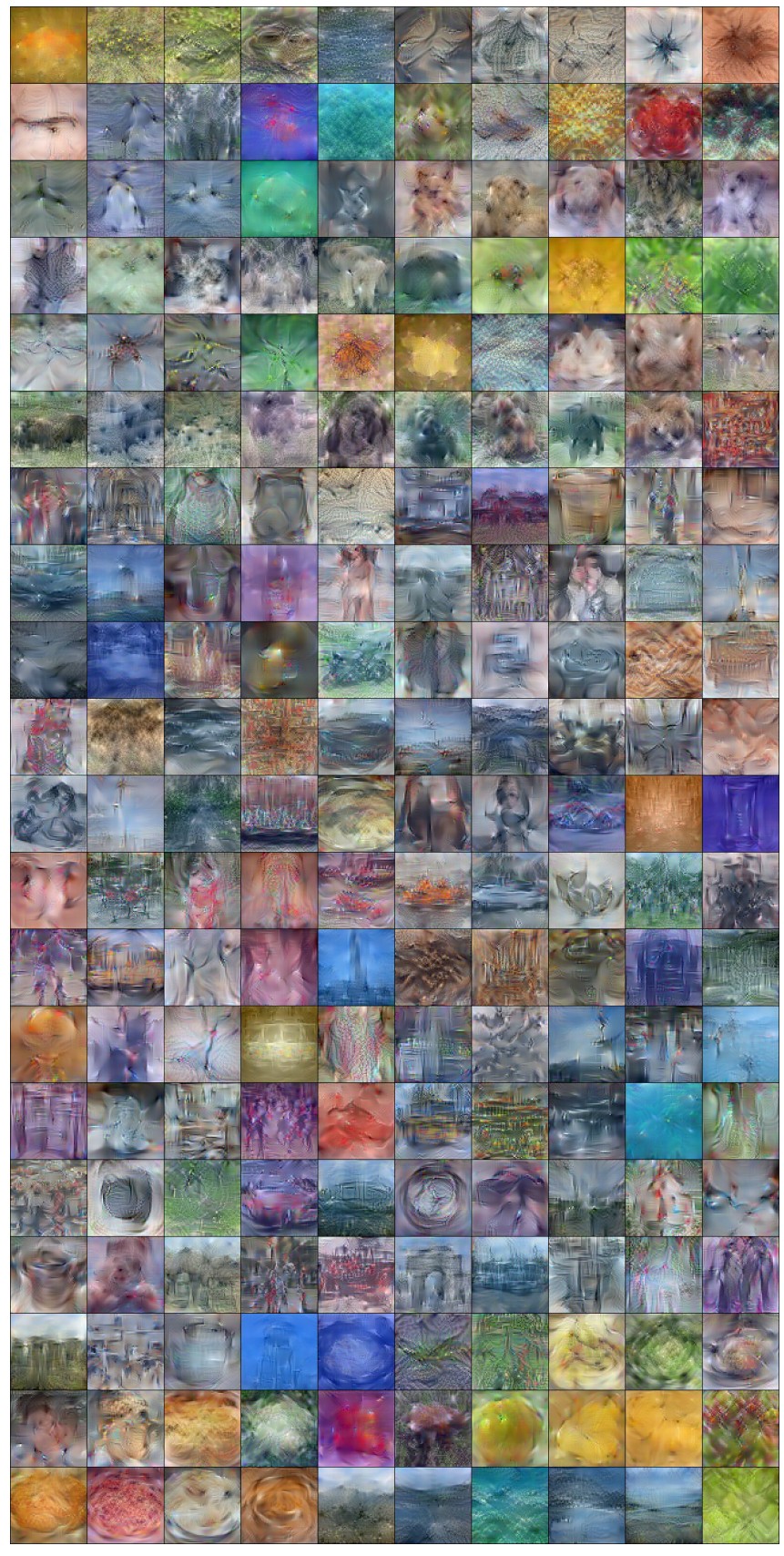

Figure 9: Distilled images of CIFAR-10 IPC10

