# OpenReview forum: "Prioritize Alignment in Dataset Distillation"
_NeurIPS.cc/2024/Conference — Submitted to NeurIPS 2024_

### Official Review · Reviewer_UeUP · 2024-07-09

**Soundness:** 3
**Presentation:** 3
**Contribution:** 2
**Rating:** 5
**Confidence:** 3

**Summary:**

This paper makes the key observation that existing dataset distillation methods often introduce misaligned information during both the extraction and embedding stages, which leads to suboptimal performances. In response to this observation, the authors propose a method called Prioritize Alignment in Dataset Distillation (PAD), which aims to filter out misaligned information through two steps: 1). Pruning the target dataset based on sample difficulty according to the compression ratio, and 2) using only deep layers of the agent model during distillation to avoid encoding low-level, redundant information.

**Strengths:**

1. This paper synthesize a universal framework for existing data distillation methods by abstracting those methods into two steps: 1). Information extraction and 2) information embedding. Furthermore, it identifies a common theme of information misalignment in both steps. This observation enhances the understanding of current limitations as well as provides a clear direction for future research.
2. The method presented in this paper effectively combines known conclusion from two distinct areas of research: data selection and representation learning. By leveraging established principles from both domains, the provide improvements to existing data distillation methods. More importantly, their analysis builds on developing an understanding of data distillation and the underlying mechanism: 1) small datasets require simple data 2) large datasets do not benefit from low-level information/features.

**Weaknesses:**

1. The proposed method’s filtering of information extraction is supported experiments shown in Figure 2. However, in practice, the method introduces two sets of hyper parameters - initial ratio and data addition epoch. The sensitivity to these hyperparameters (especially AEE shown in Table c) relative to the incremental performance gain presents a challenge, as running AEE can be complex and time-consuming (involves retraining the agent). This sensitivity and the associated tuning complexity could hinder its practical adoption in larger-scale datasets.
2. The proposed method is adapted from the DATM framework with modifications at enhancing information alignment. However, the performance improves over DATM (shown in Table 1 and in the cross-architecture generalization results in Table 3) are not significant. This marginal improvement raises concerns about the practical value of the proposed changes, as they may not justify the added complexity.

**Questions:**

1. How is data selection applied in Figure 2 (b)? In the original work, DM[1] uses untrained networks (random initialized network to embed features). In your experiment. are you using untrained agent models but applying data selection score (EL2N) from a differently pre-trained model? Overall, I am confused about how the EL2N is generated? Are you replicating the score from [2] and apply it in all your methods or are you recomputing it based on different distillation architectures?
2. Can the same analysis not applied to more recent SOTA methods such as SRe2L[3], which also involved a component similar to DM [1]? Would the same information alignments trick works as well?

[1] Bo Zhao and Hakan Bilen. Dataset condensation with distribution matching [arXiv:2110.04181](https://arxiv.org/abs/2110.04181)

[2] Mansheej Paul, Surya Ganguli, and Gintare Karolina Dziugaite. Deep learning on a data diet: 364 Finding important examples early in training. [arXiv:2107.07075](https://arxiv.org/abs/2107.07075)

[3] Zeyuan Yin, Erix Xing, and Zhiqiang Shen. Squeeze, Recover and Relabel: Dataset Condensation at ImageNet Scale From A New Perspective [arXiv:2306.13092](https://arxiv.org/abs/2306.13092)

**Limitations:**

Yes, the authors adequately addressed the limitations of their work.

---

> ### Author Rebuttal · Authors · 2024-08-06
>
> We sincerely thank the reviewer UeUP for valuable feedback. We make responses as follows.
>
> **W1: PAD introduces two new sets of parameters, which add to the complexity of tuning**
>
> Thanks for raising this concern. We would like to make the following clarifications:
>
> - They don't need to be changed according to the target datasets. Through experiments, we find that setting IR=75% and AEE=40 generalizes well across various datasets. In all experiments reported in the paper, we use these settings by default except for changing AEE to 20 on CIFAR-10.
> - Generally, 75% ~ 80% for IR and 20 ~ 40 for AEE are good settings as the performance doesn't change too much within these ranges. We show ablation results on the other two benchmarks in the tables below:
>
> **CIFAR-100, IPC10**    (notion: initial ratio = IR, addition end epoch = AEE)
>
>    | IR   | AEE=20 | AEE=30 | AEE=40   |
>    | ---- | ------ | ------ | -------- |
>    | 50%  | 47.0   | 46.9   | 47.1     |
>    | 75%  | 47.5   | 47.4   | **47.8** |
>    | 80%  | 47.4   | 47.6   | 47.7     |
>
> **Tiny ImageNet, IPC1**
>
> | IR   | AEE=20 | AEE=30 | AEE=40   |
> | ---- | ------ | ------ | -------- |
> | 50%  | 16.9   | 17.2   | 17.4     |
> | 75%  | 17.3   | 17.1   | **17.7** |
> | 80%  | 17.2   | 17.5   | 17.6     |
>
> ***
>
> **W2: PAD's improvement may not justify the added complexity**
>
> Thanks for the comment. We would like to clarify that PAD doesn't add complexity to DATM from the following three perspectives.
>
> - **Expert training:** With our difficulty measurer and data scheduler, PAD filters out misaligned information for all IPCs in one set of expert trajectories. This means we only need to train expert trajectory **once**, instead of training experts according to IPCs like in DATM. We list the comparison of expert training costs in the table below:
>
> | IPC   | DATM     | PAD             |
> | ----- | -------- | --------------- |
> | 1     | ~4 hrs.  | ~6 hrs.         |
> | 10    | ~5 hrs.  | 0 hrs. (shared) |
> | 500   | ~7 hrs.  | 0 hrs. (shared) |
> | Total | ~16 hrs. | ~6 hrs.         |
>
> - **Trajectory matching:** The matching loss is computed on fewer model parameters since we filter out part of model parameters that introduce misaligned information. In this case, the cost of distillation also doesn't increase. The comparison of running time of PAD with that of DATM is shown below. The distillation costs of these two methods are very close.
>
>   | IPC  | DATM      | PAD       |
>   | ---- | --------- | --------- |
>   | 50   | 8.3 hrs.  | 8.1 hrs.  |
>   | 1000 | 11.1 hrs. | 11.2 hrs. |
>
> - **Hyper-parameter tuning:** PAD introduces new hyper-parameters such as AEE and IR. However, these parameters are not costly to tune as we explained above.
>
> ***
>
> **Q1: How is data selection applied in Figure 2 (b)? Are you using untrained agent models but applying data selection score (EL2N) from a differently pre-trained model? Are you replicating the score from the paper and apply it in all your methods or are you recomputing it based on different distillation architectures?**
>
> Thanks for the questions. We would like to elaborate the process as follows:
>
> 1. Before the distillation begins, we use an untrained ResNet-18 to compute EL2N scores (this is the default setting of EL2N) to evaluate the difficulty of the data sample.
> 2. Then, after we filter out misaligned data points according to their scores, only reserved difficulty-matched data are used for the distribution matching.
>
> In other words,
>
> - We didn't modify the workflow of DM except now DM can only select samples from the pruned dataset during the distillation.
> - For the EL2N score, we replicate it from the paper and use it for all our methods instead of recomputing it on different architectures.
> - The implementation of computing EL2N is based on **DeepCore**.
>
> ***
>
> **Q2: Can the same analysis be applied to more recent SOTA methods such as $SRe^2L$**
>
> Thanks for the question. $SRe^2L$[8] is an excellent work on DD that proposes a "squeeze, recover, label" process to decouple previous bi-level optimization. It achieves great success on large-resolution benchmarks such as ImageNet-1k. To show PAD's compatibility, we apply PAD in the *squeeze* stage of SRe2L. Please see the experiment results and analysis below:
>
> - **Setting:** We carry out data selection to filter information extraction in the *squeeze* stage. For simplicity, experiments are done on CIFAR-100. All parameters are the same as the original paper. The model used is ResNet-18.
>
> - **Results:** Please see the table below
>
>   | IPC  | SRe2L | SRe2L + PAD(FIEX)     |
>   | ---- | ----- | --------------------- |
>   | 1    | 25.4  | 26.7 ($\uparrow$ 1.3) |
>   | 10   | 28.2  | 29.3 ($\uparrow$ 1.1) |
>   | 100  | 57.2  | 57.9 ($\uparrow$ 0.7) |
>
> - **Analysis:** After applying PAD to filter out misaligned information extracted in the *squeeze* stage, the performance of $SRe^2L$ improves on both small and large IPC settings. This further validates our hypothesis that filtering misaligned information is effective.
>
> - **Conclusion:** Filtering misaligned information can improve $SRe^2L$. It further supports that PAD is beneficial to matching-based methods for performance improvement.
>
> **We will add the discussion of $SRe^2L$ in the revision as follows:**
>
> - section 6, paragraph 5, "$SRe^2L$ introduces a "squeeze, recover, relabel" procedure that decouples previous bi-level optimization and achieves success on high-resolution settings with lower computational costs." (short version)
>
> **We will add the comparison with $SRe^2L$ in the revision as follows:**
>
> - section 5.5, a new section to discuss the compatibility of PAD on other DC/DM-based methods, add
>   "We implement the Filtering Information Extraction (FIEX) on $SRe^2L$ which involves a component similar to DM. As shown in Table 5, compared with the baseline, filtering misaligned information brings remarkable improvements on both small and large IPC settings." (short version)

---

> ### Author Response · Authors · 2024-08-11
> **To save reviewer's time, we put a summary of rebuttal**
>
> Dear Reviewer UeUP,
>
> Thanks so much again for the time and effort in our work. Considering the limited time available and to save the reviewer's time, we summarized our responses here.
>
> **Reviewer UeUP:**
>
> 1. \[**New hyper-parameters introduced**\]:
>
>    **Response:**
>    - We explain that these two hyper-parameters don't need to be changed according to different datasets, so they are not hard to tune.
>    - We provide experimental results on other benchmarks to show that AEE=20~40 and IR=75%~80% are good settings and can be generalized well.
>
> 2. \[**PAD adds more complexity**\]:
>
>    **Response:**
>    - We explain why PAD **doesn't** add any complexity from trajectory training, distillation, and hyper-parameter tuning perspectives.
>    - We provide overhead comparisons with DATM in trajectory training and distillation to further demonstrate that PAD is efficient.
>
> 3. \[**Data selection in Figure 2(b)**\]:
>
>    **Response:**
>    - We provide detailed steps on how to apply data selection in DM.
>    - The data selection is done before the distillation starts. Misaligned samples are pruned and won't be used later.
>    - We replicate the EL2N score in the paper and use default settings to compute the score.
>    - The implementation is based on Deepcore.
>
> 4. \[**Discussion of SRe2L**\]:
>
>    **Response:**
>    - We summarize the work of SRe2L and discuss the differences between PAD and SRe2L.
>    - We apply PAD on SRe2L and provide experimental results to show that PAD can bring remarkable improvements to SRe2L.
>    - We promise to include a discussion of SRe2L and the experiment in the revision.
>
>
> Since the discussion stage is already halfway through, may I know if our rebuttal addresses the concerns? If there are further concerns or questions, we are more than happy to address them. Thanks again for taking the time to review our work and provide insightful comments.
>
> Best wishes,
>
> Authors

---

> > ### Comment · Reviewer_UeUP · 2024-08-11
> > **Thank you!**
> >
> > Thank you for the detailed response and follow-up. All my concerns have been addressed. Authors clarification has demonstrated that the proposed method can be applied to more SOTA methods without introducing complicated hyperprameter tuning or computation overhead.  I would like to keep my score of acceptance!

---

> > > ### Author Response · Authors · 2024-08-12
> > > **We appreciate the support!**
> > >
> > > Thanks very much for *supporting* and *accepting* our work. We are happy that we have *addressed* your concern that hyper-parameter tuning *is not complicated*. We are also glad to see that PAD *can be applied* to more recent SOTA methods. Your constructive feedback is very *helpful* for us to improve our work.
> > >
> > > If there are any further questions, we are more than willing to provide a detailed explanation. Thank you!

---

### Official Review · Reviewer_emvM · 2024-07-12

**Soundness:** 3
**Presentation:** 3
**Contribution:** 2
**Rating:** 5
**Confidence:** 3

**Summary:**

This paper proposes to study the information misalignment problem in dataset distillation.
It proposes two basic pruning strategies: (1) learn the synthetic data with easy real samples first, and gradually change to harder samples, and (2) only match deep layers of the network during trajectory matching. The proposed method could enhance the current method in a wide range of dataset settings.

**Strengths:**

- The paper is well-written.
- The motivation is reasonable.
- The idea of using a scheduler to dynamically adjust the real sample difficulty is smart.

**Weaknesses:**

1. The experimental observations to support the two strategies (Information Extraction and Information Embedding) involving Figure 2 and 3, is not sufficient. Experiments on a wider range of datasets and more pruning ratios could rationalize the method. And a comparison of discarding deeper-layer parameters is missing.
2. The wide existence of difficulty-aware dataset distillation could **potentially** weaken the contribution. Some discussion is appreciable:
```
[1] Prune Then Distill: Dataset Distillation with Importance Sampling
[2] Distill Gold from Massive Ores: Efficient Dataset Distillation via Critical Samples Selection
[3] (DATM) Towards Lossless Dataset Distillation via Difficulty-Aligned Trajectory Matching
[4] On the Diversity and Realism of Distilled Dataset: An Efficient Dataset Distillation Paradigm
```

**Questions:**

1. Line 125-127: is there any experimental comparison to support the conclusion?
2. Is it possible to combine PAD with DATM, since they do not conflict?
3. The ablation study in Table 3(b) is weak. Conducting the experiments on other datasets may help.

**Limitations:**

The authors have discussed the limitations.

---

> ### Author Rebuttal · Authors · 2024-08-06
>
> We sincerely thank the reviewer emvM for valuable feedbacks. We make responses as follows.
>
> **W1: Experiments to support the two strategies are not sufficient**
>
> Thanks for the comment. We provide more results as follows (***will be included in the revision***)
>
> - **Settings:**
>
>   Dataset: CIFAR-10
>
>   Arch: ConvNetD3.
>
>   IPC 1/10: remove hard samples.
>
>   IPC 500: remove easy samples.
>
>   Other parameters are the same.
>
>   Init: real images.
>
> - **Results: Pls ref to Author Rebuttal Table 1~6**.
>
> - **Analyses:**
>
>   - **FIEX:**
>
>     - **Small IPCs:**
>
>       DC/DM: the performance consistently exceeds the baseline within 50% of removal.
>
>       MTT: removing up to 30% hard samples helps improve the performance.
>
>     - **Large IPCs:** removing up to 20% easy samples improves the distillation performance in large IPC settings.
>
>     - **Percentage Range:** The average percentage of data removed that exceeds baseline is nearly 40%, indicating misaligned information widely exists during the information extraction stage.
>
>   - **FIEM:**
>
>     - Both small and large IPCs benefit from discarding shallow-layer parameters.
>     - Discarding smaller ratios of shallow-layer parameters is more effective on small IPCs.
>     - Discarding larger ratios of shallow-layer parameters is more effective on large IPCs.
>
>
> - **Conclusion:** Two strategies are effective in improving performances.
>
> **CIFAR-100 is still running**. Will report when finished.
>
> ---
>
> **W2: The comparison of discarding deeper-layer parameters is missing.**
>
> Thanks for the comment. Here are results of discarding deep-layer parameters (***will be included in the revision***):
>
> - **Settings:**
>
>   Dataset: CIFAR-10
>
>   Arch: ConvNetD3
>
>   Discard ratios: 25%, 50%, 75%.
>
> - **Results:** **Please refer to Author Rebuttal Table 8**
>
> - **Analysis:** Discarding deep-layer parameters **significantly reduces** the distillation performance. This indicates that useful information is mainly distilled by deep-layer parameters.
>
> - **Conclusion:** Deep-layer parameters are more important than shallow-layer parameters.
>
> ---
>
> **W3: Discussion of other difficulty-aware DD**
>
> Thanks for the advice. We provide thorough analysis as follows (***will be put into the revision, pls ref to Author Rebuttal Revision for more details***).
>
> **Prune-then-Distill**[5] propose to prune the target dataset with data selection methods before the distillation. For each IPC, only data with high EL2N score are then used for distillation since they are recognized being **important**.
>
> - **Differences:**
>   - We use EL2N score to evaluate the **difficulty** of training examples, to filter out misaligned information. In practice, we propose to reserve samples with high EL2N scores in large IPC settings but reserving samples with low EL2N scores in small IPC settings, which turned out to be effective to all matching-based distillation methods.
>
>  **BLiP**[4] proposes part of data samples in the dataset are redundant. It designs a data utility indicator to evaluate if samples are **useful** for the distillation given an IPC setting, then samples with low utility are pruned.
>   - **Differences:**
>     - We find a few samples of target dataset are actually harmful for the distillation since they will provide over-hard or over-easy information. So we filter samples according to their difficulty measured by the EL2N score.
>     - The data scheduler allows our method to only buffer expert trajectories once for a dataset, which can be utilized by arbitrary IPC settings.
>
> **RDED**[6] proposes to extract image patches directly from the real dataset and rearrange key ones according to their scores, instead of synthesizing new images.
>
> - **Differences:**
>   - Our method focus on addressing the information misalignment problem for matching-based distillation methods, which will generate new synthetic samples during the distillation.
>
> **DATM**[7] is the first to find that the difficulty of information embedded into the synthetic data should be aligned with the IPC setting.
>
>   - **Differences:**
>     - DATM only aligns the information by controlling the matching range of training trajectories, which is only effective for methods based on matching trajectory.
>     - We find out misaligned information exists in the *Information Extraction* stage and *Information Embedding* stage. By filtering out misaligned information, our method outperforms DATM on every benchmark.
>
> ---
>
> **Q1: Experiments to support Line 125-127**
>
> Thanks for the comment. We report experimental verification as follows(***will be included in the revision***):
>
> - **Setting:**
>
>   Dataset: CIFAR-10
>
>   IPC: 500
>
> - **Results: Please refer to Author Rebuttal Table 9**
>
> - **Analysis:** Remove easy samples in one operation performs better. This supports our conclusion that after being trained on the full dataset for some epochs, letting the model focus on learning hard information is more effective for the distillation in large IPC cases.
>
> ---
>
> **Q2: Can combine PAD with DATM?**
>
> Yes, it is. As DATM is the SOTA trajectory matching method, to verify the effectiveness of PAD, our implementation is based on DATM. As shown in Table 1, PAD outperforms DATM in every setting.
>
> Moreover, PAD can also be combined with other matching-based algorithms such as DC and DM (please refer to Author Rebuttal Table 1~6).
>
> ---
>
> **Q3: More ablation studies in Table 3(b)**
>
> Thanks for your advice. We add ablation on CIFAR-100 in the revision. Here are the results:
>
> - **Settings:**
>   - Dataset: CIFAR-100
>   - IPC: 50
> - **Results: Pls ref to Author Rebuttal Table 10**
>
> - **Analyses:** Two filtering modules both bring improvements. FIEM brings more improvements.
>
>
> - **Conclusion:** Both modules are effective.
>
> **Tiny-ImageNet is still running**. Will report when finishes.

---

> ### Author Response · Authors · 2024-08-11
> **To save reviewer's time, we put a summary of rebuttal**
>
> Dear Reviewer emvM,
>
> Thanks so much again for the time and effort in our work. Considering the limited time available and to save the reviewer's time, we summarized our responses here.
>
> 1. \[**More explanations and experiments on Figure 2 and 3** \]:
>
>    **Response:**
>    - We offer results on more IPCs and test ratios as required in Table 1~6.
>    - New results further support our conclusion that misaligned information exists in matching-based DD, and our filtering strategy is effective.
>
> 2. \[**Discussion of listed works**\]:
>
>    **Response:**
>    - We discuss all mentioned works in details, including *Prune-then-Distill*, *BLiP*, *DATM*, and *RDED*.
>    - We provide thorough analyses of the differences between PAD and these works.
>    - We promise to add discussions of these works in the revision.
>
> 3. \[**Experimental results for Line 125-127**\]:
>
>    **Response:**
>    - We provide comparisons in Table 9 as required.
>    - Through experimental comparisons, our conclusion that directly removing easy samples during trajectory training is more effective has been further supported.
>
> 4. \[**Combine PAD with DATM**\]:
>
>    **Response:**
>    - We answer the question that PAD and DATM can be combined and we built PAD upon DATM.
>    - In Table 1 of the submission, PAD consistently improves DATM in all settings, further supporting our conclusion.
>
> 5. \[**More ablation studies**\]:
>
>    **Response:**
>    - We provide more ablation experiments on CIFAR-100 in Table 10 as required.
>    - The results match our conclusion: both filtering information extraction and filtering information embedding improve the performance.
>
>
> Since the discussion stage is already halfway through, may I know if our rebuttal addresses the concerns? If there are further concerns or questions, we are more than happy to address them. Thanks again for taking the time to review our work and provide insightful comments.
>
> Best Regards,
>
> Authors

---

> > ### Comment · Reviewer_emvM · 2024-08-11
> >
> > Thanks for the response and my concerns are mostly addressed. I appreciate the authors' contribution and would raise my rating to 5 since the revision meets the acceptance bar (but rejection is not that bad).
> >
> > Besides, I suggest the authors **concisely** and clearly state the novelty according to W3 response in the revision to distinguish it from other work. And there are more contemporary works that the authors could consider discussing in the final version such as [1].
> >
> > [1] SelMatch: Effectively Scaling Up Dataset Distillation via Selection-Based Initialization and Partial Updates by Trajectory Matching, ICML'24

---

> > > ### Author Response · Authors · 2024-08-11
> > > **Follow-up Discussion with Reviewer emvM**
> > >
> > > Thanks so much for the support. We appreciate the constructive feedback.
> > >
> > >
> > >
> > > Below, we provide a concise summary of our novelty:
> > >
> > > - We discover that misaligned information exists in existing matching-based DD methods. Such information comes from the mismatch between IPC capacity and actual information difficulty.
> > > - We propose to filter out misaligned information from two perspectives:
> > >   - *Information Extraction*: We conduct data selection to add/remove hard/easy samples during different phases of trajectory training.
> > >   - *Information Embedding*: We conduct parameter selection to remove shallow-layer parameters according to different IPCs during trajectory matching.
> > > - PAD can also be applied to other methods based on DC or DM.
> > >
> > > We will point out our novelty in the revision more clearly as suggested. Thanks for the reminder.
> > >
> > >
> > >
> > > Thanks for introducing an excellent work, **SelMatch**. We provide a thorough discussion as follows:
> > >
> > > - **SelMatch** points out the limitation of MTT being less effective in the large IPC setting. It proposes selection-based initialization and partial update to improve the coverage of hard and diverse patterns. SelMatch achieves remarkable success in scaling MTT on large IPCs.
> > >
> > >   **Differences:**
> > >   - **Data Perspective:** SelMatch uses all samples of the target dataset for trajectory training. PAD finds that a few samples of the dataset are actually harmful for the distillation since they will provide overly hard or overly easy information. So, PAD adds hard and removes easy samples during trajectory training to reduce the overly hard and overly easy information extracted at different training phases.
> > >   - **Trajectory Perspective:** SelMatch uses all model parameters for trajectory matching. PAD aligns the difficulty of information distilled by different model parameters with each IPC capacity. It reduces low-level basic information distilled from shallow-layer parameters and improves the distillation performance.
> > >
> > > In the revision, we will add the discussion of SelMatch as follows:
> > >
> > > - section 6, paragraph 3, "**SelMatch** proposes selection-based initialization for synthetic images to increase the coverage of hard patterns. It also employs partial update that updates part of synthetic images and keeps the rest unchanged to maintain diverse patterns. **SelMatch** successfully scales MTT as the IPC increases."
> > >
> > > We hope our response resolves your concerns. We are more than willing to answer any further questions you may have. Your support is appreciated and helps us improve our work.

---

### Official Review · Reviewer_xUdN · 2024-07-12

**Soundness:** 2
**Presentation:** 3
**Contribution:** 3
**Rating:** 6
**Confidence:** 4

**Summary:**

The authors claim that existing data distillation methods introduce misaligned information, so they propose Prioritize Alignment in Dataset Distillation (PAD). PAD prunes the target dataset and uses only deep layers of the agent model to perform the distillation, achieving state-of-the-art performance.

**Strengths:**

1. The paper is somewhat well-written and mostly easy to follow. And the tables/figures are well-demonstrated.

2. The authors analyze the misaligned information from two perspectives and propose method.

3. PAD achieves improvements on various benchmarks, achieving state-of-the-art performance.

**Weaknesses:**

The performance gains brought by the method proposed by the authors are subtle and limited, potentially attributable to other explanations. For instance, as mentioned in [1], discarding original data in certain ways, or even randomly, can yield minor performance improvements under different IPC conditions. Or tricks mentioned in [2].

The trend changes in Figure 2 are not pronounced, and there are even instances where the trends contradict the explanations. Could additional test ratios or test IPCs be included to validate the findings?

[1] Distill Gold from Massive Ores: Efficient Dataset Distillation via Critical Samples Selection.
[2] Data Distillation Can Be Like Vodka: Distilling More Times For Better Quality

**Questions:**

In Figure 2, Figure 3, and Table 5, the baselines do not all align with those in Table 1 or the results presented in the original paper. Is there an explanation I have missed? Please clarify this discrepancy, as I am unable to evaluate the correctness of "Misaligned Information Extracted" without understanding the reason for this variation.

**Limitations:**

Due to the limitation of computing resources, the authors only validated their method’s effectiveness on DATM, DM, and DC.

---

> ### Author Rebuttal · Authors · 2024-08-06
>
> We thank the reviewer xUdN for the feedback. We make responses as follows.
>
> **W1: Performance gains are limited and may have other explanations**
>
> Thanks for the question. In Table 1, we achieve 11 SOTAs out of 12 settings. Moreover, PAD can be generalized to methods based on matching gradients (DC[1]) and distributions (DM[2]), further showing the effectiveness of filtering out misaligned information.
>
> To clarify our differences between "PDD" and "BLiP", we make analyses and experimental verification as follows (***will be put into the revision, please refer to Author Rebuttal Revision for more details***):
>
> - **PDD**[3] progressively synthesizes multiple sets of images, to capture the training dynamics of different stages. It achieves success in enhancing existing DD methods.
>   **Differences:**
>   - **Data Perspective:** PDD uses all samples of the target dataset. PAD filters out samples that contain misaligned information according to the IPC setting, and they won't be used during the entire distillation process.
>   - **Trajectory Perspective:**  PAD aligns the difficulty of information distilled by different model parameters with each IPC capacity and improves the distillation performance.
> - **BLiP**[4] proposes that it is not necessary to use all samples of the target dataset to perform the distillation since they are redundant. It greatly improves the distillation efficiency by pruning a large percentage of redundant data.
>   **Differences:**
>   - **Data Perspective:**  BLiP proposes a data utility indicator to evaluate if samples are 'useful' given an IPC setting, then samples with low utility are pruned. We find that a few samples of the target dataset are actually harmful for the distillation since they will provide over-hard or over-easy information. So, PAD filters data samples according to difficulty measured by EL2N scores.
>   - **Trajectory Preparation Perspective:** For trajectory-matching-based methods, BLiP needs to train expert trajectories case by case due to different pruning ratios. Our data scheduler allows our method to only buffer expert trajectories once, which can be utilized by arbitrary IPC settings.
>
> **Experimental Comparison with BLiP:**
>
> - **Settings:** For a fair comparison, we only compare the improvement brought by our data selection module (FIEX) with BLiP. Experiments are done on CIFAR-10 and we use MTT[5] as the distillation method. The data pruning ratios and IPCs tested are the same as BLiP.
>
> - **Results: Pls ref to Author Rebuttal Table 7**
>
> - **Analyses:**
>
>   - PAD brings better performance improvements on IPC1/10/50.
>   - Under the given data-dropping ratios, PAD's improvements over BLiP get larger as the IPC increases.
>
> - **Conclusion:** Difficulty misalignment between IPC and real data used is more harmful. PAD's data selection module is more effective in removing such misaligned information.
>
> ***
>
> **W2: Figure 2 needs to be explained**
>
> Thanks for the comment. In Figure 2, we show a group of ablation results to demonstrate that removing misaligned information from the information extraction step helps improve the performance. We make additional clarifications as follows:
>
> - The percentage of easy(hard) samples to remove is not always the higher the better for large(small) IPCs. When the removing ratio crosses a certain value, we may lose too much information to achieve a good performance. For example, the performance of trajectory matching on IPC500 is below baseline when removing more than 20% of easy samples. This explains why some points are below the baseline.
> - We find the percentage of easy(hard) samples that are not aligned with large(small) IPCs resides in a **large range** (around 40%, please refer to Tables 1,2,3 in the Author Rebuttal). This means misaligned information widely exists in previous DD and our proposed solution is effective.
>
> ***
>
> **W3: Ablation under more IPCs and reduction ratios in Figure 2.**
>
> Thanks for the advice. More ablation studies are reported as follows:
>
> - **Settings:**
>
>   Dataset: CIFAR-10
>
>   Architecture: ConvNetD3.
>   IPC 1/10: remove various ratios of hard samples.
>   IPC 500: remove various ratios of easy samples.
>
>   Initialization: real images.
>
>   Other parameters: same as the default.
>
> - **Results:** Please refer to **Author Rebuttal Table 1~3**.
>
> - **Analyses:**
>
>   - **Small IPCs:** Removing part of hard samples helps improve the distillation performance.
>
>   - **Large IPCs:** Removing part of easy samples improves the distillation performance.
>   - **Percentage Range:** The average percentage of data removed that gives better performance than the baseline is nearly 40%, indicating that misaligned information is a common issue for existing DD, and our filtering strategy is effective.
>
>
> - **Conclusion:** Misaligned information indeed exists and filtering it out can alleviate the negative effect.
>
> ***
>
> **Q1: Results of DC and DM in Figure 2, 3 don't match Table 1**
>
> We are sorry for the confusion. We elaborate on the reasons as follows:
>
> - In the DC and DM, there are two ways of synthetic image initialization, random gaussian noise and images from the real dataset. In the current version, we initialize the synthetic data with random noise.
> - We used smaller inner loop and outer loop steps in previous experiments for convenience.
>
> Thanks for the reminder. To avoid confusion, we adjust our experiments with real image initialization and official hyper-parameter settings. New results are reported as follows:
>
> - **Settings:**
>
>   Init: real images
>
>   Other parameters: same as the default.
>
> - **Results:** Please refer to **Author Rebuttal Table 1~6**.
>
> - **Analysis:**  As can be observed, the trend still matches our previous experiments. Removing difficulty misaligned information distilled by shallow-layer model parameters according to IPCs achieves better performances.
>
> - **Conclusion:** Misaligned information exists in the matching-based methods. Filtering it out brings improvements.

---

> ### Author Response · Authors · 2024-08-11
> **To save reviewer's time, we put a summary of rebuttal**
>
> Dear Reviewer xUdN,
>
> Thanks so much again for the time and effort in our work. Considering the limited time available and to save the reviewer's time, we summarize our responses here.
>
> 1. \[**Other explanations to performance gains**\]:
>
>    **Response**:
>    - PAD achieves SOTAs on 11 out of 12 settings.
>    - We provide a detailed discussion of the two mentioned works, BLiP and PDD.
>    - We also provide experimental comparisons between PAD and BLiP in Table 7: PAD is more effective than BLiP on MTT.
>
> 2. \[**More explanations and experiments on Figure 2**\]:
>
>    **Response**:
>    - We offer more clarifications on the trend of Figure 2 and explain why some points could be below the baseline.
>    - According to the requirement, we add more IPCs and test ratios and listed all results in Author Rebuttal Table 1~3.
>    - Results still support our conclusion that misaligned information exists in existing matching-based DD, and removing it improves the performance.
>
> 3. \[**Figure 2,3 results and Table 1 don't match**\]:
>
>    **Response**:
>    - We explain reasons why these results don't match: we used noise for synthetic image initialization and we reduced the inner and outer loop steps.
>    - We rerun all experiments with the default hyper-parameter setting and fix the results in Figure 2,3. Updated results are shown in Author Rebuttal Table 1~6. The trend still matches our previous results.
>    - We will update all results in the revision.
>
>
> Since the discussion stage is already halfway through, may I know if our rebuttal addresses the concerns? If there are further concerns or questions, we are more than happy to address them. Thanks again for taking the time to review our work and provide insightful comments.
>
> Best Regards,
>
> Authors

---

> > ### Comment · Reviewer_xUdN · 2024-08-12
> > **Official Comment by xUdN**
> >
> > Thanks for the author's efforts and thorough clarifications. The author has provided detailed supplementary explanations and experiments and the rebuttal has addressed most of my concerns. I‘d like to increase my score.

---

> > > ### Author Response · Authors · 2024-08-12
> > > **We appreciate the support**
> > >
> > > Thanks very much for the support. We are happy to see that we have addressed your concerns. Your valuable feedback is very helpful for us in improving our work.
> > >
> > > If you have any further questions, we are more than happy to provide a detailed explanation. Thanks again for accepting our work.

---

### Author Rebuttal · Authors · 2024-08-06

We sincerely thank all reviewers for their valuable feedback, which is very important to further improve our work.
We begin by making the following responses about results of additional experiments and revision of the paper, and later to allow more space for responses to each reviewer.

### Tables

**Table 1: Filtering Information Extraction in Matching Gradients**

| CIFAR-10 | baseline | 5%   | 10%      | 15%  | 20%      | 25%      | 30%  | 50%  |
| - | - | - | - | - | - | - | - | - |
| IPC1     | 27.8     | 28.0 | 28.4     | 28.5 | **29.1** | 28.8     | 28.1 | 27.9 |
| IPC10    | 44.7     | 45.2 | 45.5     | 45.7 | 46.1     | **46.3** | 45.3 | 44.7 |
| IPC500   | 70.8     | 71.7 | **71.9** | 71.2 | 71.4     | 70.3     | 69.8 | 67.1 |

**Table 2: Filtering Information Extraction in Matching Distributions**

| CIFAR-10 | baseline | 5%   | 10%  | 15%      | 20%      | 25%  | 30%  | 50%      |
| - | - | - | - | - | - | - | - | - |
| IPC1     | 26.4     | 26.5 | 27.1 | 27.3     | 27.9     | 28.2 | 28.5 | **29.2** |
| IPC10    | 48.4     | 48.6 | 48.9 | 49.7     | **50.3** | 49.6 | 49.2 | 48.5     |
| IPC500   | 75.1     | 75.6 | 76.2 | **76.3** | 75.8     | 75.3 | 74.6 | 74.2     |

**Table 3: Filtering Information Extraction in Matching Trajectories**

| CIFAR-10 | baseline | 5%   | 10%      | 15%      | 20%      | 25%  | 30%  | 50%  |
| - | - | - | - | - | - | - | - | - |
| IPC1     | 46.4     | 46.9 | 47.1     | 47.3     | **47.6** | 47.2 | 47.0 | 46.7 |
| IPC10    | 66.5     | 66.7 | 67.2     | **67.4** | 67.2     | 67.3 | 66.8 | 65.4 |
| IPC500   | 83.5     | 83.6 | **84.3** | 83.9     | 83.5     | 83.2 | 82.7 | 81.1 |

**Table 4: Filtering Information Embedding in Matching Gradients**

| CIFAR-10 | baseline | 12.5% | 25%      | 50%  | 62.5% | 75%      |
| - | - | - | - | - | - | - |
| IPC10    | 44.6     | 44.8  | **45.2** | 44.7 | 44.1  | 43.8     |
| IPC500   | 72.2     | 72.3  | 72.5     | 72.8 | 73.2  | **73.4** |

**Table 5: Filtering Information Embedding in Matching Distributions**

| CIFAR-10 | baseline | 12.5% | 25%      | 50%  | 62.5% | 75%      |
| - | - | - | - | - | - | - |
| IPC10    | 48.9     | 49.1  | **49.5** | 49.1 | 48.5  | 48.3     |
| IPC500   | 75.1     | 75.2  | 75.5     | 75.9 | 76.2  | **76.3** |

**Table 6: Filtering Information Embedding in Matching Trajectories**

| CIFAR-10 | baseline | 12.5% | 25%      | 50%  | 62.5 | 75%      |
| - | - | - | - | - | - | - |
| IPC10    | 66.8     | 67.1  | **67.2** | 66.9 | 66.2 | 65.5     |
| IPC500   | 83.5     | 83.7  | 83.8     | 84.2 | 84.3 | **84.5** |

***

**Table 7: Comparison between BLiP and PAD on MTT**

Notion: The left in the bracket denotes the improvement over MTT, and the right denotes the percentage of real data used for distillation.

| IPC  | PAD (Ours)           | BLiP             |
| - | - | - |
| 1    | 46.8 (**+0.6**, 80%) | 46.3 (+0.2, 80%) |
| 10   | 66.5 (**+1.1**, 90%) | 65.7 (+0.4, 90%) |
| 50   | 73.0 (**+1.4**, 95%) | 72.0 (+0.4, 95%) |

***

**Table 8: Performances of discarding deep-layer parameters for distillation**

| IPC  | PAD (w/o data selection) | 25%            | 50%            | 75%             |
| - | - | - | - | - |
| 1    | 46.9                     | 44.1      (-2.8) | 43.2      (-3.7) | 41.8       (-5.1) |
| 10   | 66.9                     | 62.2      (-4.7) | 57.7      (-2.8) | 51.1      (-15.7) |
| 50   | 76.1                     | 69.2      (-6.9) | 66.5      (-9.6) | 58.3      (-17.8) |

***

**Table 9: Comparison between direct removal and gradual removal of easy samples.**

| Strategy        | CIFAR-10 IPC500 | CIFAR100 IPC50   |
| - | - | - |
| Gradual remove  | 84.2            | 55.6             |
| Directly remove | 84.6       (+0.4) | 55.9        (+0.3) |

***

**Table 10: Ablation of modules on CIFAR-100 IPC50**

| FIEX | FIEM | Accuracy |
| - | - | - |
|      |      | 55.0     |
| ✓ | | 55.2     |
||✓| 55.6|
|✓|✓| 55.9|



### Revision

**We thank the Reviewer xUdN for adding two other related works. We will add the discussion and experiment results in the revision as follows:**

- section 6, paragraph 3, add "**BLiP**\[4\] discovers the issue of data redundancy in previous distillation..."
- section 6, paragraph 3, add "**PDD**\[3\] identifies the change of learned pattern complexity at different training stages..."
- section 5.4, add "We compare FIEX with **BLiP**\[4\] on MTT. As shown in Table 5, FIEX in PAD performs better on IPC1/10/50..."



**We thank Reviewer emvM for introducing three other difficulty-aware distillation works. In the revision, we will add the discussion of these papers in the related work as follows:**

- in section 6, paragraph 3, "**BLiP**\[4\] and **Prune-then-Distil**\[5\] discover the issue of data redundancy..."
- in section 6, paragraph 3, "**PDD**\[3\] identifies the change of learned pattern complexity at different training stages... " ( *will present its results in Table 1* in the revision)
- in section 6, paragraph 5, "**RDED**\[6\] proposes a computationally efficient DD method that doesn't require synthetic image..." (*will present its results in Table 1* in the revision)
- in section 5.4, we add experimental comparisons and analysis between PAD and **BLiP**\[4\].





### **References**

[1] *Dataset Condensation with Gradient Matching*, ICLR 2020

[2] *Dataset Condensation with Distribution Matching*, CVPR 2021

[3] *Data Distillation Can Be Like Vodka: Distilling More Times For Better Quality*, ICLR 2024

[4] *Distill Gold from Massive Ores: Efficient Dataset Distillation via Critical Samples Selection*, ECCV 2024

[5] *Prune Then Distill: Dataset Distillation with Importance Sampling*, ICASSP 2023

[6] *On the Diversity and Realism of Distilled Dataset: An Efficient Dataset Distillation Paradigm*, CVPR2024

[7] *Towards Lossless Dataset Distillation via Difficulty-Aligned Trajectory Matching*, ICLR 2024

[8] *Squeeze, Recover and Relabel: Dataset Condensation at ImageNet Scale From A New Perspective*, NeuIPS 2023

---

### Author Response · Authors · 2024-08-12
**Summary of Discussion and Plans for Revision**

Dear Reviewers and ACs,

We express our gratitude to all reviewers and ACs for taking so much time in our work. We appreciate all the advice and feedback from reviewers, which are very helpful for us to improve our work.

Here, we list all the updates we will make in the revision:

- **Text:** We will add discussions of the following works:
  - **BLiP:** in section 6, paragraph 3
  - **PDD:** in section 6, paragraph 3
  - **Prune-then-distill:** in section 6, paragraph 3
  - **RDED**: in section 6, paragraph 5
  - **SRe2L:** in section 6, paragraph 5
  - **SelMatch:** in section 6, paragraph 3
- **Experiments:** We will fix and add experimental results as follows:
  - Results in the Figure 2 and 3 will be fixed and complemented (more ratios, IPCs, and datasets).
  - Ablation studies on CIFAR-100 and Tiny-ImageNet will be added.
  - Performance of discarding deep-layer parameters will be added.
  - Comparison between directly removing easy samples and gradually removing easy samples will be added.
  - Comparison between BLiP and PAD on MTT will be added in section 5.4.
  - Results of applying PAD on SRe2L will be added in section 5.5.

Again, we thank all reviewers for supporting and accepting our work. We will continue to improve our work in the revision. If there are any further inquiries, we are more than willing to discuss them.

Best Regards,

Authors

---

### Author Response · Authors · 2024-08-13
**Promised Experimental Results of Figure 2,3 (will be included in the revision)**

Dear Reviewers and ACs,

As promised, we present abaltion results to support our two strategies, Filtering Information Extraction (FIEX) and Filtering Information Embedding (FIEM), on CIFAR-100 and Tiny-ImageNet as follows:

- **Settings:**
  - FIEX:
    - Dataset: CIFAR-100
    - IPC1: Remove hard samples
    - IPC100: Remove easy samples
  - FIEM:
    - Dataset: Tiny-ImageNet
- **Results:** Please see the tables below.

**Filtering Information Extraction**

Matching Gradients

| CIFAR-100 | baseline | 10%  | 20%      | 30%      | 50%  |
| --------- | -------- | ---- | -------- | -------- | ---- |
| IPC1      | 12.8     | 13.2 | 13.5     | **14.4** | 12.6 |
| IPC100    | 44.5     | 45.1 | **45.6** | 45.5     | 43.2 |

Matching Distributions

| CIFAR-100 | baseline | 10%  | 20%      | 30%      | 50%  |
| --------- | -------- | ---- | -------- | -------- | ---- |
| IPC1      | 11.4     | 12.1 | 12.5     | **13.2** | 12.7 |
| IPC100    | 46.2     | 46.8 | **47.1** | 46.5     | 44.9 |

Matching Trajectories

| CIFAR-100 | baseline | 10%  | 20%      | 30%      | 50%  |
| --------- | -------- | ---- | -------- | -------- | ---- |
| IPC1      | 24.3     | 24.7 | 24.8     | **25.3** | 24.1 |
| IPC100    | 49.2     | 49.6 | **50.4** | 48.9     | 47.6 |



**Filtering Information Embedding**

Matching Trajectory

| Tiny-ImageNet | baseline | 25%     | 50%  | 75%      |
| ------------- | -------- | ------- | ---- | -------- |
| IPC1          | 8.8      | **9.5** | 8.9  | 8.5      |
| IPC50         | 28.0     | 28.3    | 28.9 | **29.4** |

- **Analyses:**
  - **FIEX:**
    - On small IPCs, removing up to 30% hard samples helps improve the distillation performance;
    - On large IPCs, removing up to 20% easy samples helps improve the distillation performance.
  - **FIEM:**
    - Both small and large IPCs benefit from discarding shallow-layer parameters.
    - Discarding shallow-layer parameters is more effective as the IPC increases.

- **Conclusion:**  Two strategies are effective in improving the performance of matching-based methods.

We will add the above results in the revision. Again, we appreciate the support. If there are any other questions, please don't hesitate to discuss them with us.

Best Regards,

Authors

---

### Decision · Program_Chairs · 2024-09-25

**Decision:**

Reject

**Comment:**

This paper studies the problem of dataset distillation, that is compressing a large dataset into a smaller dataset while preserving model performance when trained on the data. While the problem setting is interesting/relevant -- the experimental evidence is weak the authors only have results on CIFAR  and TinyImageNet and even there the win relative to baseline is small. The reviewers echo similar concerns. For that reasons I vote to reject this paper.